# Effect of Manual Therapy and Splint Therapy in People with Temporomandibular Disorders: A Preliminary Study

**DOI:** 10.3390/jcm9082411

**Published:** 2020-07-28

**Authors:** Gemma Victoria Espí-López, Anna Arnal-Gómez, Alba Cuerda del Pino, José Benavent-Corai, Pilar Serra-Añó, Marta Inglés

**Affiliations:** 1Department of Physiotherapy, Faculty of Physiotherapy, University of Valencia, Gascó Oliag Street nº 5, 46010 Valencia, Spain; gemma.espi@uv.es (G.V.E.-L.); albacuerda96@gmail.com (A.C.d.P.); pilar.serra@uv.es (P.S.-A.); marta.ingles@uv.es (M.I.); 2Research Unit in Clinical Biomechanics (UBIC), University of Valencia, 46010 Valencia, Spain; 3Cavanilles Institute for Biodiversity and Evolutionary Biology, University of Valencia, 46980 Paterna, Spain; Jose.benavent.corai@gmail.com

**Keywords:** musculoskeletal manipulation, physical therapy modalities, pain, temporomandibular joint disorder, oral health

## Abstract

Background: Isolated manual therapy techniques (MT) have shown beneficial effects in patients with temporomandibular disorders (TMD) but the effect of the combination of such techniques, together with the well-stablished splint therapy (ST) remains to be elucidated. Objective: This study was conducted to ascertain whether a combined program of MT techniques, including intraoral treatment, plus traditional ST improves pain and clinical dysfunction in subjects with TMD. Methods: A preliminary trial was conducted. 16 participants were assigned to either the MT plus ST-Experimental Group (EG, n = 8) or the ST alone—Control Group (CG, n = 8). Forty-five minute sessions of combined MT techniques were performed, once a week for four weeks. Three evaluations were conducted: baseline, post-treatment, and one-month follow-up. Outcome measures were pain perception, pain pressure threshold (PPT), TMD dysfunction, and perception of change after treatment. Results: EG showed significant reduction on pain, higher PPT, significant improvement of dysfunction and significantly positive perception of change after treatment (*p* < 0.05 all). Additionally, such positive effects were maintained at follow-up with a high treatment effect (R^2^ explaining 26.6–33.2% of all variables). Conclusion: MT plus ST showed reduction on perceived pain (3 points decrease), higher PPT (of at least 1.0 kg/cm^2^), improvement of disability caused by pain (4.4 points decrease), and positive perception of change (EG: 50% felt “much improvement”), compared to ST alone.

## 1. Introduction

Temporomandibular disorders (TMD) are defined by a set of clinical signs that affect the temporomandibular joint (TMJ), the masticatory muscles, and related structures involved in the movements of the TMJ [1].

It has a prevalence of over 5% [2], and is characterized by different symptoms and signs which include, TMJ or surrounding tissues pain, generalized myofascial pain, joint noise in the form of clicking and associated with movement, decrease joint movement amplitude, functional limitations, and deviation from jaw opening [1,3]. The etiology of TMD can be multifactorial and due to multiple causes including emotional, psychological, structural, and biomechanical factors [4]. Some authors defend the relationship of TMD with cervical dysfunction [5], which may indicate they could share risk factors. Therefore, the term TMD is broad and it contains a number of disease entities [6]. Thus, in order to support evaluation and diagnosis an assessment protocol, the diagnostic criteria for temporomandibular disorders (DC/TMD) has been developed [7].

Different treatment options for TMD are currently available: (a) splint therapy (ST), which has been widely studied and its efficacy proved [3,8,9]; thus, being proposed as part of a reversible occlusal treatment that includes elimination or reduction of pain levels and frequency [3], reduction of excessive activity, restoration of symmetry of masticatory muscles tone [8], and an increase of mouth opening ability [3]; (b) pharmacologic approaches, by taking analgesics, anti-depressants, muscle relaxant, or non-steroidal anti-inflammatory drugs [10], or by intra-articular administration of medications, which has been proved to have a positive impact on the reduction of the intensity of pain [11]; (c) psychological support [3] and behavioral changes, such as reduction of stress levels, are also important factors in the management of patients suffering from TMD [12]; (d) physical therapy techniques involving, active and passive stretching, endurance exercises of involved muscles, postural exercises, and isolated manual therapy techniques (MT) that have proven to be effective in TMD treatment [13]; and (e) in severe cases, surgical procedures are sometimes applied [6].

In this regard, a relationship between TMD and cervical dysfunction (one of the major goals of MT techniques) has been highlighted by several authors, explained by the topographic arrangement between the cervical spine and the trigeminal nerve [14]. Thus, upper neck mobilization has been shown to reduce pain and improve mobility of the craniocervical region in subjects with TMD [15,16]. In line with this, it has been reported that soft tissue manipulative treatment in suboccipital trigger points [17] and in the TMJ region improve muscle function, joint movement, and pain [18]. With regard to TMJ itself, its accessory mobilization decreases pain and muscle spasms, and increases range of motion [19,20]. Moreover, myofascial release techniques on masseter and pterygoid muscles have shown a reduction in TMJ pain [21], and massage has proven to produce local analgesia and improvement of muscle function [22].

Therefore, isolated MT techniques have shown beneficial effects in patients with TMD, but the effect of the combination of such techniques, together with the well-stablished ST remains to be elucidated. Thus, a combined therapy of the currently broadly used ST, and articular/myofascial techniques applied both at the TMJ and the cervical level would allow the clinicians to address joint and myofascial pain, functional limitation and cervical mobility, all of them important variables that have an impact on people who suffer from TMD.

In the present study, we aimed to ascertain whether a program based on a combination of MT techniques, which have shown beneficial effects when applied isolated, plus the traditional treatment of ST in subjects with TMD decreases pain, improves pressure threshold, dysfunction, and positively influences perception of change on patients when compared to the application of ST alone.

## 2. Methods

### 2.1. Participants

The sample included participants who suffered from TMD, diagnosed by dentists applying the DC/TMD [7], and were recruited using a purposive sampling technique. The inclusion and exclusion criteria are shown on Table 1.

### 2.2. Study Design

A preliminary randomized controlled trial was carried out from May 2018 to August 2018. The participants were randomly allocated by a statistician to two different groups: the experimental group (EG), in which a protocol of accessory mobilization and myofascial MT techniques and ST was applied, and the control group (CG), who received ST alone. Participants belonging to the EG were treated for four weeks (one 45-min session per week). To analyze the effect of the intervention, three evaluations were carried out: at baseline (T0), post-treatment (T1), and at follow-up (i.e., one month after treatment completion) (T2).

The treatments were applied by an experienced physical therapist, with the necessary training and the study was carried out in the laboratories of the Physical Therapy Faculty of the University of XXXX. Participants provided informed consent following an explanation of the study aims and procedures before entering the study. The current preliminary clinical trial was conducted following the Consolidated Standards of Reporting Trials (CONSORT) statement [23]. The Ethics Review Board of the University of XXXX approved all the procedures (H1509658631780), which were performed in accordance with the principles of the Declaration of Helsinki of the World Medical Association and its revision in 2013. This trial was registered at www.clinicaltrials.gov (registration number: NCT03555201).

### 2.3. Randomization and Blinding

Patients were randomly assigned to the EG or CG by an assistant who did not participate in the trial. Sequentially numbered envelopes were prepared with random assignment. The assessment files were placed in sealed opaque envelopes. Another assistant from outside the study proceeded with the assignment of the treatment. Coding, analysis, and interpretation of results was done by an external assistant.

### 2.4. Interventions

Both groups used a personalized occlusal splint for all the duration of the study (8 weeks) and only the EG also received the articular and myofascial MT protocol (4 weeks), which consisted of ten different techniques as described in Appendix A. Following the intervention, subjects remained in supine position with a neutral head and neck position for two minutes [17].

Splint therapy: The splints were manufactured individually prescribed by a dentist, who made additional adjustments if necessary. It consisted of conventional occlusal splint molded with irreversible hydrocolloid (Algitex, Dentsply) for the fabrication of a Michigan-type splint with canine and protrusive guides as well as a flat occlusal surface for contact with the antagonist teeth. Patients were instructed to use their occlusal splints 12 h a day for the duration of the study [24].

MT treatment: The protocol consisted of a combination of ten techniques applied on the cervical, suboccipital, and temporomandibular areas: (1) neck accessory mobilization technique on C7 vertebra [16]; (2) mobilization neck central with posterior–anterior C5 vertebra mobilization [15]; (3) mobilization of upper neck [15]; (4) suboccipital inhibition technique for two minutes [17]; (5) suboccipital accessory mobilization technique with occipito-atlo-axoidea (OAA) thrust, a maximum of two attempts was permitted to achieve cavitation or the audible pop, as perceived by the therapist and/or the patient [17]; (6) trigger points technique on masseter, temporal, and sternocleidomastoid muscles [18]; (7) myofascial technique on masseter, pterygoid lateral, and medial muscles [21]; (8) TMJ mobilization technique [19]; and (9) temporomandibular massage [22].

### 2.5. Outcomes

#### 2.5.1. Primary Outcome

Pain: Assessed with the Visual Analogue Scale (VAS), in which the patients marked their level of pain intensity on a 10-cm horizontal line (0 = no pain to 10 = maximum pain) in relation to the last week. When pain was unilateral, only one side was registered, when bilateral, patient was asked to indicate a global score of pain. The VAS has shown high validity and reliability for the assessment of the patient’s pain intensity (CI 95% = 0.96–0.98) [25]. Participants were instructed to communicate any use of medication due to an acute increase in pain in the TMJ region during the duration of the study.

#### 2.5.2. Secondary Outcomes

Pain pressure threshold (PPT): The minimal pressure (kg/cm^2^) which induces pain was measured by pressure algometry (Wagner Instruments FDK 20). The patient was seated, and the masseter, temporal, and sternocleidomastoid muscles were assessed bilaterally, performing three measurements in each muscle, with a period of thirty seconds rest between them. The average of the three scores was obtained for analysis. The reliability of pressure algometry procedure was found to be high (ICC = 0.91 (95% confidence intervals (CI): 0.82–0.97) [26] and moderate in patients with TMD (ICC = 0.64) [27]. When testing treatment effect of PPT a principal component analyses (PCA) was performed to summarize in one variable both sides (left and right) of masseter, both sides of temporal and both sides of sternocleidomastoid muscle features. In fact, the first component in each PCA explained 90.1% for masseter, 93.1% for temporal and 96.3% for sternocleidomastoid variables.

Dysfunction Index of TMD: It was assessed by the Helkimo Index which includes the following five clinical signs and symptoms (scored with 0, 1, or 5 points each): (i) impaired range of mandibular mobility; (ii) TMJ function impairment; (iii) muscle pain; (iv) TMJ pain during palpation; and (v) pain during mandibular movement. The total score is based on the sum of the score of the five items, with 25 being the maximum score. The index classifies the individual as follows: absence of TMD signs and symptoms (0 points), mild TMD signs and symptoms (score range, 1–4 points), moderate TMD signs and symptoms (5–9 points), and serious TMD signs and symptoms (10–25 points) [28].

Change perception: Assessed at T2 with the Patient Global Impression of Change Scale (PGICS), which evaluates the change perceived by the subject after treatment. This scale has seven affirmations depicting a patient’s rating of overall improvement (1 = “very much improved”, 2 = “much improved”, 3 = “minimally improved”, 4 = “no change”, 5 = “minimally worse”, 6 = “much worse”, or 7 = “very much worse”). Therefore, the lower the score, the higher the improvement perception [29].

#### 2.5.3. Statistical Analysis

Descriptive results of continuous data were expressed as mean and standard deviation while categorical data were described as frequencies and percentages. In order to test treatment effect we applied in standardized data, a non-parametric permutational multivariate analysis of variance (PERMANOVA) with a repeated measure design that included three factors [30,31]: factor A, treatment; factor B, subjects; and factor C, time. Although PERMANOVA is robust to heterogeneity of variance in balanced designs, we tested it applying PERMDISP which is a multivariate analogue of Leven’s test [31]. Post-hoc pair-wise tests among time treatments were tested through the F-ratio applying Monte Carlo methodology (*p*-values were obtained using 150 permutations) [30]. These permutation methodologies are very useful for testing hypothesis when normality of residuals could not be tested or assumed. Cohen’s f^2^ statistic [32] was used to estimate the effect size of both “treatment” and interaction of “treatment*time” factors. All statistical analyses were performed with custom scripts implemented in R 2.15.1 statistical software (R Development Core Team 2015, R Foundation) and SAS version 9.4 (SAS Institute Inc., Cary, NC, USA). The effect size (Cohen’s *d*) [33] was calculated for comparisons where statistically significant differences were obtained within and between groups. Statistical significance was set at *p*-value < 0.05.

## 3. Results

### 3.1. Participants

Of the twenty-five prospective participants, sixteen individuals were eligible, agreed to participate, were randomly assigned to a group, and completed the study (EG, n = 8; CG, n = 8). Seven participants did not meet inclusion criteria and were excluded: three participants had surgical history in TMD area, three had previous physical therapy, and one participant had used analgesics or muscle relaxants. Two participants voluntarily abandoned the study for personal reasons. Figure 1 shows the process of participant recruitment and dropouts. No participants had adverse effects.

Therefore, sixteen patients were considered for analyses in the study, with a mean age of 29.9 (±12.4) years, being mostly women (n = 13; 81%). Participants were classified with myofascial pain (EG, n = 4; CG, n = 4) and myofascial pain with referral (EG, n = 4; CG, n = 4) according to DC/TMD. Participants of both groups had pain for at least four years, with a weekly frequency and of moderate intensity. The pain focused mostly on the jaw and cervical area. Clinical profiles of participants for both groups are shown in Table 2. They did not significantly differ (*p* > 0.05) in morphometric and pain variables, except for Body Mass Index and Head Pain Location.

### 3.2. Effect of the Treatment

When each of the factors of the study were analyzed, the treatment factor explained significantly 27.4% of the variance (F_1, 14_ = 5.4, *p* = 0.002) and the interaction between treatment*time factor explained 11.3% (F_2, 28_ = 3.1, *p* = 0.001). When considering the EG, pair-wise comparison showed significant differences between T0 and T1 (F = 13.7, *p* = 0.001) and between T0 and T2 (F = 10.3, *p* = 0.001), but not between T1 and T2 (F = 0.4, p = 0.146). These results indicate that the EG experienced a significant general improvement after the treatment in relation to the CG. Moreover, the effect of that treatment was maintained significantly throughout the time of experimentation.

The ANOVA analysis showed significant differences between groups for all measured variables at T1 and T2 (Table 3). The results of the VAS showed that participants of EG significantly improved pain at T1 (*p* = 0.001) and this improvement was maintained at T2 (*p* = 0.001), with the treatment factor explaining 33.2% (R^2^) of pain improvement with a large effect size (f^2^: 0.50). No differences were found within the CG. In relation to the differences between groups, statistical changes were found for VAS at T1 and T2, with a large effect (*p* = 0.001, Cohen’s *d* = 0.8 for both). None of the participants communicated any use of medication due to an acute increase in pain in the TMJ region for the duration of the study. 

The Helkimo Index results showed a significant decrease of 4.4 points in the EG between T0 and T1 (*p* = 0.001), so it improved from being practically in a severe level of dysfunction to a moderate one, staying at this level at T2 (*p* = 0.02). CG on the other hand showed no statistical within group differences; thus, remained at values close to severe dysfunction throughout the time of experimentation, with all participants being at moderate or severe levels. In the EG at T0 50% of the participants had a severe level of dysfunction and 50% had a moderate level, then at T1 and T2 none of the participants had a severe level, and they were all between moderate to low level of dysfunction (Table 4).

In relation to PPT (algometry), Figure 2 shows there was an improvement in the EG for the masseter, temporal, and sternocleidomastoid when considered bilaterally. However, the CG remained at similar values throughout the entire study time. In fact, the treatment factor explained at least 26.6% (R^2^) of the improvement of the algometry variable for the three muscle groups, with a moderate to large effect size depending on the muscle (f^2^: 0.36 and 0.38 for temporal and sternocleidomastoid, respectively; f^2^: 0.51 for masseter). Thus, the between groups differences were statistically significant for all the measured muscles both at T1 and T2. Algometry results for EG show significant increase for all of the muscle groups at T1: masseter muscle significantly increased at least 1.0 kg/cm^2^ (*p* = 0.001), temporal muscle showed a significant increase of at least 1.9 kg/cm^2^ (*p* = 0.001), and sternocleidomastoid muscle significantly increased at least 1.3 kg/cm^2^ (*p* = 0.001). For the CG generally at T1 and T2 results were maintained or decreased when compared with T0, and no within groups statistical differences were found.

Regarding the effect of the treatment over time, it was noted that for the EG at T1 the VAS and the Helkimo decreased significantly, and the algometry of the three muscle groups increased significantly and these improvements were maintained at T2.

Finally, PGICS scale showed that after the completion of the whole study (T2) EG perceived a greater change after treatment (CG: 4.3, SD = 0.9; EG: 2.4, SD = 1.4) and this difference between groups was significant (mean difference = 1.9, *p* = 0.005). Moreover, 50% of participants of EG felt “much improvement” after the treatment, whereas 50% of CG felt “minimally worse”.

## 4. Discussion

The current preliminary study shows that a MT-based protocol combined with ST tends to improve short- and midterm clinical parameters of pain (VAS and algometry), severity, dysfunction, and perception of change in individuals with TMD when compared to the traditional treatment based on ST alone. These results highlight that MT combined with ST can be a more effective treatment than only ST. In addition, the positive effects of MT are maintained over time, since we observed improvements even at follow up.

To the best of our knowledge, this is the first study to evaluate the potential effects of adding a protocol including several MT techniques to the traditional ST for TMD. This protocol included both accessory mobilization and myofascial techniques applied to the TMJ and the cervical region, in order to address the most relevant clinical aspects of TMD.

The results of this study showed significant improvement of pain measured by VAS in the participants of the EG compared to the CG. The EG results at T1 and T2 for VAS when compared with T0, both showed significant improvement, although not with the same intensity. Moreover, the EG experienced an improvement beyond the minimum clinically significant difference in pain severity (i.e., greater than three points) [35]. It is noteworthy that such improvement was observed both immediately after treatment and at follow-up, which may indicate that the proposed combined treatment induces positive hypoalgesic effects in the manipulated segments, which are maintained over time. This improvement may be due, on the one hand, to accessory mobilization manipulations at the cervical area, since they have been shown to be effective in inducing changes on muscles innervated by the manipulated segment and, on the other hand, due to the soft-tissue techniques applied, since they are advantageous in the management of musculoskeletal disorders [36,37].

The neuro-anatomical basis which may explain this relationship between the neck and the head may be related to the trigemino-cervical nucleus caudalis in the spinal grey matter of the spinal cord at the C1–C3 level, where there is a convergence on the nociceptive second order neurons receiving trigeminal and cervical inputs [38]. This topographic arrangement allows the interchange of nociceptive information between the cervical spine and the trigeminal nerve [14]. Therefore, individuals with prolonged suboccipital muscle tension are likely to suffer irritation of the first cervical nerve (C1), which induces the stimulation of associated sympathetic fibers. This irritation of the C1 spinal nerve is manifested by referred pain in the TMJ area and the neck [39], and vice versa; that is, stimulation of trigeminal-innervated structures evokes painful sensations in the neck [40]. Therefore, it is possible that the suboccipital techniques that were applied in our study activated segmental inhibitory pathways [41] via the trigeminal nucleus caudalis; thus, causing positive effects on pain. It appears that MT can improve pain due to the existence of an intimate functional relationship between the mandibular and the head–neck systems. Our study provides preliminary evidence that MT applied both at the cervical spine and the masticatory area may be beneficial in decreasing pain in patients suffering from TMD.

Noteworthy, PPT improved significantly for the three studied muscle groups only in the EG, while it did not improve in the CG. Thus, the combined treatment applied to the EG was more effective in improving the PPT than the traditional ST; thus, suggesting that it may have improved the nociceptive stimulus on painful tissue [42] related to specific areas of masseter, temporal, and sternocleidomastoid muscles. Individuals who suffer TMD exhibit reduced blood flow to the masticatory muscles due to vasoconstriction stemming from muscle hyperactivity. Consequently, the transport of nutrients and metabolites is impeded, which can lead to the build-up of by-products; thereby, triggering pain [43]. Therefore, the observed improvements may be due to the fact that soft tissue MT stimulates local blood flow and re-establishes normal muscle status in individuals with myofascial mandibular pain and muscle spasm [44,45]. Furthermore, by including accessory mobilization techniques such as suboccipital inhibition and OAA manipulation, the PPT of the masseter and temporal muscles significantly increased, which is in agreement with previous studies [17].

Regarding the improvement of the dysfunction assessed by the Helkimo Index, taking into account that this index assesses clinical signs related to both pain and mobility, our results indicate that the combination of MT techniques with ST allowed to improve the two main limitations of TMD. Although the use of ST has shown positive effects in subjects with TMD in previous studies [46] by adding the MT techniques, clinical improvements seem to be increased.

Considering that the Helkimo Index assesses range of mandibular mobility, pain during mandibular movement, and function impairment, among other symptoms, the improvements achieved in our study may be explained by to the MT protocol, rather than the ST. The applied MT treatment included techniques, such as non-thrust joint mobilization, which have been found to improve extensibility of non-contractile tissue and increase range of motion while decreasing pain and disability via peripheral, spinal, and supraspinal mechanisms [20]. Moreover, previous studies have also found significant reduction in the symptoms of dysfunction in TMJ after applying intraoral myofascial techniques on masseter and pterygoids [21].

In line with our results, other authors who have applied isolated manual or soft tissue techniques have also reported clinical improvements in individuals with TMD. In this regard, a previous study showed that soft-tissue techniques were more effective than ST to improve mandibular range of motion in subjects with TMD [47]. Furthermore, Cuccia et al. reported that both MT (using joint and soft tissue techniques) and ST were favorable for the assessed variables (VAS, range of motion, and maximal mouth opening), but the MT group required less pharmacological medication than ST group (*p* < 0.001) [48]. Taken together, these results and ours indicate that the addition of a MT protocol to the traditional ST induces a greater impact on the improvement of the signs and symptoms of TMD.

Another important aspect was the participant’s perception after the treatment, which can reveal relevant information of the beneficial effects of a given therapy. To date, most studies on TMD have proven the effectiveness of treatment from the physical condition dimension, while, in the present study, the perception of change was also taken into account. Our results showed significant differences between groups in relation to the patient’s belief about the efficacy of treatment, with the EG patients perceiving between minimum and much improvement, and the CG perceiving between no change and minimally worse. Although the PGICS scale has been widely used in chronic pain clinical trials [49], it has not been used in chronic TMD. However, it is important to identify appropriate measures to assess functional changes [50] among patients with chronic pain related to TMD receiving treatment, since a perception of improvement after the treatment may reduce stress levels, and therefore, improve how they manage their condition. Furthermore, associations of PGICS ratings with other outcomes measures (e.g., depression, sleep, and vitality) have been demonstrated in other conditions [51] indicating the relationship between pain and other important health aspects which may be also considered as future research lines in TMD patients.

As for limitations, the main one is the small sample size which is a usual situation in physical therapy. Nevertheless, we performed an ad hoc statistical design (PERMANOVA). Among its advantages is that it is robust to non-normal data and allows controlling the emergence of false positives. Another limitation of the study is the source of participants, which was mainly from dental practices, which biases the selection of participants. Thus, our results must be taken with caution and further studies with larger sample sizes, which would allow subgroups according to disorders, would be recommended to confirm our promising results. One of the strengths of this study is that, although there are many previous studies that evaluate MT in pain, which we consider needed, we also included the level of dysfunction and the change that the subject perceived after being treated. In addition, reproducible, simple, and gentle techniques applied in the cervical region and in the TMJ have been included.

## 5. Conclusions

The results of the present study suggest that a combined protocol based on MT plus ST tend to improve pain, pain-induced disability, and the patients’ self-perception of change in individuals with TMD. Moreover, such positive effects are maintained after one-month follow-up. These results reinforce the effectiveness of a multimodal treatment provided by a multidisciplinary team collaborating in the therapeutic approach for TMD patients.

## Figures and Tables

**Figure 1 jcm-09-02411-f001:**
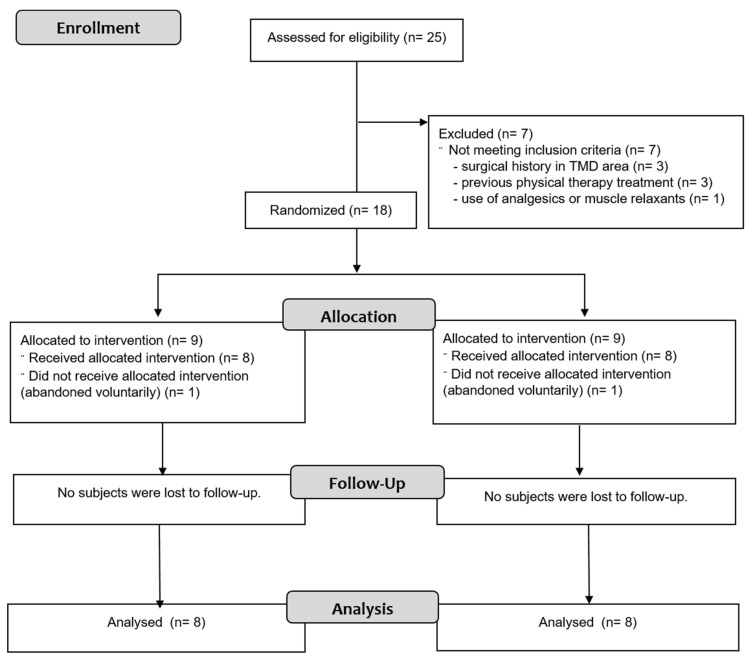
Flowchart According to the Consolidated Standards of Reporting Trials (CONSORT) Statement.

**Figure 2 jcm-09-02411-f002:**
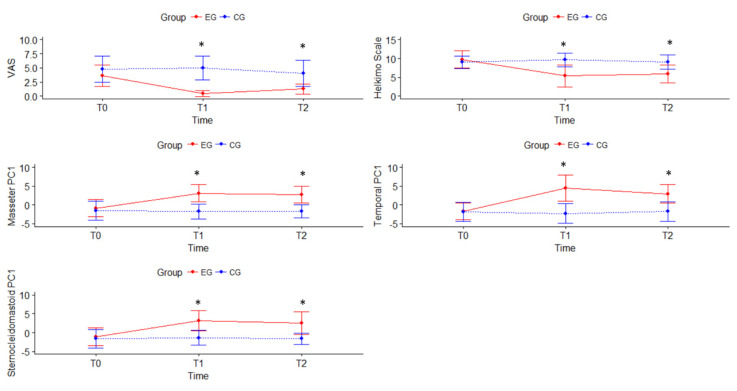
Results of Visual Analogue Scale, Helkimo, and Algometry for Experimental Group (EG) and Control Group (CG). T0 = baseline measurement; T1 = post-treatment measurement; T2 = 1-month follow-up measurement. Algometry shows the Principal Component Analysis (PCA) combining in one variable both sides (left and right) of masseter muscle (MasseterPC1), of temporal muscle (TemporalPC1) and sternocleidomastoid muscle (SternoceidomastoidPC1). * significant differences (*p* < 0.05) between groups.

**Table 1 jcm-09-02411-t001:** Participants inclusion and exclusion criteria.

Inclusion Criteria	Exclusion Criteria
- Aged 18 to 65. - Diagnosed with at least mild TMD signs and symptoms according to Helkimo Index. - Diagnosed with pain disorders according to DC/TMD: myalgia and myofascial pain (with history of pain in masticatory structure, and/or modified by jaw movement function or parafunction; and confirmation of pain in masticatory muscles with palpation) [7].	- Systemic, rheumatic, or central nervous system diseases. - Surgical history in TMD area - Previous physical therapy treatments (last 3 months). - Diagnosed with other orofacial or TMJ disk disorders. - Vertebral artery compromise test. - Cerebrovascular disorders. - Use of analgesics or muscle relaxants at least 24 h before assessments and during the treatment period. - Use of splint 1 month before the start of the study.

Abbreviations: TMD: temporomandibular disorders; DC/TMD: diagnostic criteria for temporomandibular disorders; TMJ: temporomandibular joint.

**Table 2 jcm-09-02411-t002:** Participants’ Demographic and Baseline Clinical Characteristics.

Variables	CG (n = 8)	EG (n = 8)	*p-*Value
Age (years)	29.8 ± 14.6	30.0 ± 11.6	0.967
Gender (%male/%female)	0/100	37.5/62.5	N.A.
BMI	20.5 ± 2.2	23.0 ± 2.0	0.031 *
VAS ^a^	4.8 ± 2.3	3.6 ± 1.9	0.238
Helkimo Scale	9.0 ± 1.7	9.8 ± 2.3	0.407
Algometry ^b^			
Right Masseter	4.7 ± 0.8	4.9 ± 0.8	0.741
Left Masseter	4.5 ± 0.9	4.8 ± 0.7	0.452
Right Temporal	4.5 ± 0.8	4.6 ± 0.6	0.859
Left Temporal	4.9 ± 1.0	4.9 ± 0.9	0.837
Right Sternocleidomastoid	3.8 ± 0.8	4.0 ± 0.7	0.640
Left Sternocleidomastoid	4.0 ± 0.8	4.2 ± 0.8	0.657
Pain characteristics			
When it started (years)	6.8 ± 4.0	4.5 ± 2.4	0.202
Intensity (VAS last 2 weeks)	5.3 ± 1.8	6.0 ± 1.9	0.294
Frequency (%monthly/%weekly/%daily)	12.5/50/37.5	0/75/25	0.270
Severity (%mild/%moderate/%severe) ^c^	12.5/87.5/0	25/50/25	0.176
Pain Location (%never/%sometimes/%frequently)			
Head	50/25/25	0/75/25	0.020 *
Jaw	0/37.5/62.5	0/50/50	0.317
Cervical	0/62.5/37.5	12.5/37.5/50	0.319
Shoulders	75/25/0	62.5/25/12.5	0.582

Data are shown as mean ± standard deviation. Abbreviations: CG = control group; EG = experimental group; BMI = Body Mass Index; VAS = Visual Analogue Scale; ^a^ VAS of the last week; ^b^ Algometry shows the measurements for both sides (kg/cm^2^); and ^c^ VAS levels: 1–4 (mild pain), 5–6 (moderate pain), and 7–10 (severe pain) [34]. * *p* < 0.05.

**Table 3 jcm-09-02411-t003:** Measurements of VAS, Helkimo, Algometry, and comparison analysis.

Variables	Groups	T0	T1	T2	Mean Difference (95%CI); Effect Size (*d*)				
					Within-Group Differences			Between-Groups Differences	
					T0–T1	T0–T2	T1–T2	T1	T2
VAS ^a^	CG	4.8 ± 2.3	5.0 ± 2.1	4.0 ± 2.3	−0.2 (−0.6 to 2.1)	0.8 (−1.1 to 0.6)	1.0 (−0.3 to 2.3)	−4.5 (−6.3 to −2.7); *d* = 0.8	−2.8 (−4.7 to −0.8); *d* = 0.8
	EG	3.6 ± 1.9	0.5 ± 0.5 *	1.3 ± 0.9 *^,†^	3.1 (1.7 to 4.4); *d* = 0.3	2.4 (1.1 to 3.7); *d* = 0.4	−0,8 (−1.3 to 0.2); *d* = 0.5		
Helkimo Index	CG	9.0 ± 1.7	9.6 ± 1.8	9.1 ± 1.9	−0.6 (−1.5 to 0.3)	−0.1 (−0.9 to 0.7)	0.5 (−0.1 to 0.9)	−4.2 (−6.9 to −1.6); *d* = 1.2	−3.2 (−5.6 to −0.9); *d* = 1.1
	EG	9.8 ± 2.3	5.4 ± 2.9 *	5.9 ± 2.4 *	4.4 (2.7 to 6.0); *d* = 0.4	3.9 (1.5 to 6.3); *d* = 0.4	−0.5 (−2.1 to 1.1)		
Algometry ^b^									
Right Masseter	CG	4.7 ± 0.8	4.8 ± 0.8	4.8 ± 0.8	−0.1 (−0.3 to 0.2)	−0.1 (−0.4 to 0.2)	−0.0 (−0.2 to 0.1)	1.1 (0.3 to 1.9); *d* = 0.4	1.2 (0.4 to 2.1); *d* = 0.4
	EG	4.9 ± 0.8	5.9 ± 0.7 *	6.0 ± 0.9 *	−1.0 (−1.4 to −0.6); *d* = 0.4	−1.1 (−1.7 to −0.7) *d* = 0.4	−0.1 (−0.4 to −0.1)		
Left Masseter	CG	4.5 ± 0.9	4.4 ± 0.7	4.4 ± 0.6	0.1 (−0.2 to 0.5)	0.1 (−0.3 to 0.6)	− 0.0 (−0.2 to 0.1)	1.9 (1.1 to 2.6); *d* = 0.4	1.5 (0.8 to 2.2); *d* = 0.3
	EG	4.8 ± 0.7	6.3 ± 0.8 *	5.9 ± 0.7 *	−1.5 (−1.8 to −1.0); *d* = 0.4	−1.1 (−1.6 to −0.6); *d* = 0.4	0.4 (−0.2 to 0.9)		
Right Temporal	CG	4.5 ± 0.8	4.8 ± 0.9	4.8 ± 0.9	−0.3 (−0.7 to 0.3)	−0.3 (−0.7 to 0.1)	−0.0 (−0.4 to 0.2)	1.7 (0.7 to 2.9); *d* = 0.5	1.4 (0.5 to 2.3); *d* = 0.4)
	EG	4.6 ± 0.6	6.5 ± 1.0 *	6.2 ± 0.8 *^,^^†^	−1.9 (−2.7 to −1.2); *d* = 0.3	−1.6 (−2.3 to −0.9): *d* = 0.3	0.3 (−0.0 to 0.7); *d* = 0.5		
Left Temporal	CG	4.9 ± 1.0	4.4 ± 0.8	4.7 ± 0.9	0.4 (−0.1 to 1.0)	0.2 (−0.2 to 0.5)	−0.3 (−0.7 to 0.2)	2.5 (1.4 to 3.5); *d* = 0.5	1.6 (0.6 to 2.5); *d* = 0.4
	EG	4.9 ± 0.9	6.9 ± 1.1 *	6.3 ± 0.8 *^,^^†^	−2.0 (−2.8 to −1.2); *d* = 0.4	−1.4 (−1.8 to 0.9); *d* = 0.4	0.6 (0.1 to 1.1); *d* = 0.5		
Right SCM	CG	3.8 ± 0.8	3.9 ± 0.6	3.9 ± 0.5	−0.1 (−0.5 to 0.2)	−0.1 (0.5 to 0.4)	0.0 (−0.2 to 0.4)	1.6 (0.7 to 2.3); *d* = 0.4	1.5 (0.6 to 2.3); *d* = 0.4
	EG	4.0 ± 0.7	5.5 ± 0.9 *	5.4 ± 1.0 *	−1.5 (−1.9 to −1.1); *d* = 0.4	−1.4 (−1.8 to −0.8); *d* = 0.4	0.1 (−0.4 to 0.6)		
Left SCM	CG	4.0 ± 0.8	4.0 ± 0.6	3.9 ± 0.5	−0.0 (−0.5 to 0.4)	0.1 (−0.3 to 0.4)	0.1 (−0.1 to 0.3)	1.5 (0.6 to 2.2); *d* = 0.4	1.3 (0.4 to 2.2); *d* = 0.4
	EG	4.2 ± 0.8	5.5 ± 0.9 *	5.2 ± 1.0 *	−1.3 (−1.8 to −0.8); *d* = 0.4	−1.0 (−1.8 to −0.3); *d* = 0.4	0.3 (−0.4 to 0.9)		

Data are shown as mean ± standard deviation. Abbreviations: CG = control group; EG = experimental group; T0 = baseline measurement; T1 = post-treatment measurement; T2 = 1-month follow-up measurement; SCM = Sternocleidomastoid; ^a^ VAS of the last week; ^b^ algometry shows the measurements for both sides (kg/cm^2^); * significant differences (*p* < 0.05) compared to T0; ^†^ significant differences (*p* < 0.05) compared to T1; and bold indicates significant differences (*p* < 0.05) between groups. Cohen’s d effect size was calculated for comparisons where statistically significant differences were obtained.

**Table 4 jcm-09-02411-t004:** Helkimo Index and Patient Global Impression of Change Scale.

	CG (n = 8)			EG (n = 8)		
	T0	T1	T2	T0	T1	T2
Helkimo Index, frequency, (%)						
Absence (0 points)	0 (0%)	0 (0%)	0 (0%)	0 (0%)	0 (0%)	0 (0%)
Mild (1–4 points)	0 (0%)	0 (0%)	0 (0%)	0 (0%)	4 (50%)	3 (37.5%)
Moderate (5–9 points)	6 (75%)	5 (62.5%)	5 (62.5%)	4 (50%)	4 (50%	5 (62.5%)
Severe (10–25 points)	2 (25%)	3 (37.5%)	3 (37.5%)	4 (50%)	0 (0%)	0 (0%)
PGICS, frequency, (%)						
1 = “very much improved”	0 (0%)			2 (25%)		
2 = “much improved”	0 (0%)			4 (50%)		
3 = “minimally improved”	2 (25%)			0 (0%)		
4 = “no change”	2 (25%)			1 (12.5%)		
5 = “minimally worse”	4 (50%)			1 (12.5%)		
6 = “much worse”	0 (0%)			0 (0%)		
7 = “very much worse”	0 (0%)			0 (0%)		

Abbreviations: CG = control group; EG = experimental group; T0 = baseline measurement; T1 = post-treatment measurement; T2 = 1-month follow-up measurement; and PGICS: Patient Global Impression of Change Scale.

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
