# Peer review of "Effect of Manual Therapy and Splint Therapy in People with Temporomandibular Disorders: A Preliminary Study"

_jcm, 2020, doi:10.3390/jcm9082411_

Round 1
Reviewer 1 Report
This article investigates the efficacy of combining splint therapy with manual therapy for the treatment of TMD.
To improve the significance of the findings of this study, the authors should clarify some facts throughout the manuscript and provide additional information to enhance the understanding of the readers and clarify the results.
1. The etiology of TMD is known to be complicated however only cervical disorders is described. Please add more information on the other possible mechanisms and etiologies of TMD.
2. Medication is also a part of TMD treatment including others that are not described in the Introduction. Please add information.
3. In the Methods the study group is poorly defined as "who suffered from TMD, diagnosed by dentists", also as "Diagnosed with myogenic TMD or
myofascial pain according to Helkimo Index".
The international consortium for the research of TMD recommends to follow the DC/TMD to produce reliable study results.
Currently there is no detailed explanation of the exact diagnostic process that was followed to define the TMD group also patients of various TMD subgroups have been combined into a single group that could that affected the results.
4. Additional description of the splint that was used in the study is necessary as type, design, and material.
5. Why was the VAS (last 2 weeks) not used as baseline data instead of the VAS that was from an undescribed period.
6. What is the criteria to differentiate mild/moderate/severe pain levels?
7. Was the mean value from the left and right muscles used for analysis?
In cases when the patient had unilateral TMD pain, how would have this affected the results?
8. Was all the other treatment protocols identical between the 2 groups? Was medication given in case the patient showed an acute increase in pain?
9. Was both splint and manual therapy stopped after 1 month and before the final evaluation time point?
10. Data regarding the post-treatment pain characteristics should be presented.
Author Response
In relation to the manuscript with Title “Effect of manual therapy and splint therapy in people with temporomandibular disorders: a preliminary study” we would like to thank the reviewer for the comments on our manuscript. For sure the aspects that have been changed due to the comments will help improve the understanding of our research and its impact. Each comment has been addressed individually in this separate letter and changes in the manuscript have been highlighted with yellow colour so that they can be easily identified and reviewed.
Point 1: The etiology of TMD is known to be complicated however only cervical disorders is described. Please add more information on the other possible mechanisms and etiologies of TMD.
Response 1: We concur with the reviewer that TMD’s etiology is complicated, therefore we have added more mechanisms to clarify there can be different possible causes.
Manuscript:
- Lines 44 and 45 “The etiology of TMD can be multifactorial and due to multiple causes including emotional, psychological, structural, and biomechanical factors.”
Point 2: Medication is also a part of TMD treatment including others that are not described in the Introduction. Please add information.
Response 2: Following the reviewers comment, information about medication options has been added. Moreover, also other possible interventions have been included.
Manuscript:
- Lines 54 and 55 “b) Pharmacologic approaches, by taking analgesics, anti-depressants, muscle relaxant, or non-steroidal anti-inflammatory drugs,”
- Line 61 “e) In severe cases, surgical procedures are sometimes applied”
Point 3: In the Methods the study group is poorly defined as "who suffered from TMD, diagnosed by dentists", also as "Diagnosed with myogenic TMD or myofascial pain according to Helkimo Index".
The international consortium for the research of TMD recommends to follow the DC/TMD to produce reliable study results.
Currently there is no detailed explanation of the exact diagnostic process that was followed to define the TMD group also patients of various TMD subgroups have been combined into a single group that could that affected the results.
Response 3: We thank the reviewer for this suggestion which has helped define our study group. In order to clarify the diagnostic process of the participants, more information has been added. In the Introduction section, the Diagnostic Criteria for Temporomandibular Disorders (DC/TMD) has been included.
In Methods section, the inclusion of participant according to DC/TMD criteria has been added, both in the text and in Table 1.
In Results section the characteristics of the participants following the DC/TMD criteria has been added. As stated in lines 190 and 191 clinical profiles of participants for both groups did not significantly differ.
In relation to the fact that various TMD subgroups have been combined into a single group and that this could affect the results, we concur with the reviewer, and therefore, it has been added as a limitation.
Manuscript:
- Lines 47 to 49 “Therefore, the term TMD is broad and it contains a number of disease entities. Thus, in order to support evaluation and diagnosis an assessment protocol, the Diagnostic Criteria for Temporomandibular Disorders (DC/TMD) has been developed.”
- Lines 84 and 85 “applying the DC/TMD,”
- Line 87, Table 1: “Diagnosed with pain disorders according to DC/TMD.”
- Line 186 and 187: “Participants were classified with myofascial pain (EG, n=4; CG, n=4) and myofascial pain with referral (EG, n=4; CG, n=4) according to DC/TMD.”
- Lines 343 to 345: “Thus, our results must be taken with caution and further studies with larger sample sizes, which would allow subgroups according to disorders, would be recommended to confirm our promising results.”
Point 4: Additional description of the splint that was used in the study is necessary as type, design, and material.
Response 4: Following the reviewer’s recommendation, more information about the splint used in the study has been added.
Manuscript:
- Lines 117 to 119: “It consisted of conventional occlusal splint moulded with irreversible hydrocolloid for the fabrication of a Michigan-type splint with canine and protrusive guides as well as a flat occlusal surface for contact with the antagonist teeth.”
Point 5: Why was the VAS (last 2 weeks) not used as baseline data instead of the VAS that was from an undescribed period.
Response 5: We concur with the reviewer that the information given previously in the manuscript was not clear in this regard. Therefore, information about the VAS has been clarified in the tables. The VAS that was used for comparisons was the one in relation to the last week in order to be able to compare the data at baseline (T0), after the 4-week treatment (T1) and after one-month follow-up (T2).
Manuscript:
- Line 134: “in relation to the last week”.
- Line 194 Table 2 Participants’ Demographic and Baseline Clinical Characteristics: “VAS of the last week”
- Lines 214 Table 3 Measurements of VAS, Helkimo, Algometry and ANOVA analysis: “VAS of the last week”
Point 6: What is the criteria to differentiate mild/moderate/severe pain levels?
Response 6: In relation to the criteria to define pain levels with VAS, more information has been added to the manuscript.
Manuscript:
- Lines 195 and 196 Table 2: VAS levels: 1-4 (mild pain), 5-6 (moderate pain), and 7-10 (severe pain) (Jensen, M.P., Smith, D.G., Ehde, D.M., & Robinsin, L.R. (2001). Pain site and the effects of amputation pain: further clarification of the meaning of mild, moderate, and severe pain. Pain, 91(3), 317-322.)
Point 7: Was the mean value from the left and right muscles used for analysis?
In cases when the patient had unilateral TMD pain, how would have this affected the results?
Response 7: When pain (measured with VAS) was unilateral, only one side was registered. When it was bilateral, the patient was asked to indicate a score which globally represented the suffered pain. This information has been added to the manuscript in the Methods section.
Manuscript:
- Lines 134 and 135: “When pain was unilateral, only one side was registered, when bilateral, patient was asked to indicate a global score of pain”.
Point 8: Was all the other treatment protocols identical between the 2 groups? Was medication given in case the patient showed an acute increase in pain?
Response 8: The treatment protocol was identical between the 2 groups regarding the splint therapy, and only the experimental group had manual therapy protocol added. We have added more information regarding time points of the protocol to clarify this procedure.
In case of any use of medication due to an acute increase in pain in the TMJ region, participants were instructed to communicate it. None of the participants communicated any use of medication in this regard. This information has been added in the Methods and Results section of the manuscript.
Manuscript:
- Lines 112 to 114 “Both groups used a personalised occlusal splint for all the duration of the study (8 weeks) and only the EG also received the articular and myofascial MT protocol (4 weeks), which consisted of ten different techniques as described in Appendix.”
- Lines 137 and 138: “Participants were instructed to communicate any use of medication due to an acute increase in pain in the TMJ region during the duration of the study.”
- Lines 208 and 210: “None of the participants communicated any use of medication due to an acute increase in pain in the TMJ region for the duration of the study.”
Point 9: Was both splint and manual therapy stopped after 1 month and before the final evaluation time point?
Response 9: Thank you for this comment, which is in line with the one before, and both will help to better understand the procedure of the study. The splint therapy was used by the two groups for all the duration of the study (8-weeks) and the manual therapy protocol was conducted for the first 4 weeks only on the experimental group. This information has been clarified in the manuscript.
Manuscript:
- Lines 112 and 113: “Both groups used a personalised occlusal splint for all the duration of the study (8 weeks) and only the EG also received the articular and myofascial MT protocol for 4 weeks,”
Point 10: Data regarding the post-treatment pain characteristics should be presented.
Response 10: Pain measurements obtained with the VAS (in the last week), and with the algometry are presented at Table 3 for baseline (T0), after the 4-week treatment (T1) and after on month follow-up (T2). We registered more pain characteristics at baseline in order to know clinical situation of participants and we thought interesting to show them. However, no pain characteristics were registered after treatment. This comment will be taken into consideration for future research.
Manuscript:
- Line 211 Table 3. Measurements of VAS, Helkimo, Algometry and ANOVA analysis.

Reviewer 2 Report
I would personally thank the authors to have carried on this interesting study in the controversial world of the TMD’s treatment. The work is clear, concise and well written. Nevertheless, some parts of the text must be reformulated as we must always consider the nature of this study (preliminary study).
Abstract
Please reformulate the objective by highlighting the importance of the tongue training strengthening.
In the conclusions please give more effort on the results of the study.
Introduction
The Introduction would benefit from rewriting in some parts.
I would start by citing the classification of the temporo-mandibular disorders of Dworkin et al. (2014). Moreover, this classification should be followed by the authors to better define the characteristics of the samples involved in the study. Indeed, one of the most controversial point regarding the TMD is the patient’s classification.
Line 54-62: I would reformulate these sentences by focusing on the possible anatomical-functional relations linking TMD with the cervical area and then specify the effects of the manual therapy.
Methods
2.1 Participants - Table I- inclusion criteria- Joint noises during mouth opening
I would reformulate specifying the type classifying it according to the Dworkin classification (if possible).
2.4 Interventions
Personalized occlusal splint: I would only specify if there are any specific characteristics of the splint that according to the authors should be taken into consideration or not.
Discussion
Considering the low impact of the present work as a preliminary study I would start the discussion reformulating the first sentence and making it “softer”.
Line 267: I would spend some more words explaining how the manual therapy can cause positive effects on pain, as here lies the core of the study.
Line 282: How do the author can explain why in the CG has been reported “minimally worse” in 50% and “no change” in 25% of the patients?
Conclusions
As for the first paragraph of the Discussion section I would suggest the authors to reconsider the conclusion in a “softer way”.
Tables and Figures:
I found all the tables clear and well structured. I would suggest the authors to pay attention in Table 3 and 4 there are still some “,” instead of “.”.
I personally found the Appendix interesting as additional value to the study.
Author Response
In relation to the manuscript with Title “Effect of manual therapy and splint therapy in people with temporomandibular disorders: a preliminary study” we would like to thank the reviewer for the comments on our manuscript. For sure the aspects that have been changed due to the comments will help improve the understanding of our research and its impact. Each comment has been addressed individually in this separate letter and changes in the manuscript have been highlighted with yellow colour so that they can be easily identified and reviewed.
Point 1: Abstract. Please reformulate the objective by highlighting the importance of the tongue training strengthening. In the conclusions please give more effort on the results of the study.
Response 1: Although no specific tongue strengthening training has been applied in the protocol, the intraoral techniques used are also related to the tongue and other TMJ related structures, so these have been highlighted in the objective of the Abstract as suggested by the reviewer. Also the conclusions of the abstract have been more related to the results of the study.
Manuscript:
- Lines 20 and 21: “including intraoral treatment,”
- Lines 30 to 33: “MT plus ST showed reduction on perceived pain (3 points decrease), higher PPT (of at least 2.3 kg/cm2), improvement of disability caused by pain (4.4 points decrease) and positive perception of change (EG: 50% felt “much improvement”), compared to ST alone.”
Point 2: Introduction. The Introduction would benefit from rewriting in some parts. I would start by citing the classification of the temporo-mandibular disorders of Dworkin et al. (2014). Moreover, this classification should be followed by the authors to better define the characteristics of the samples involved in the study. Indeed, one of the most controversial point regarding the TMD is the patient’s classification.
Response 2: We thank the reviewer for suggesting the reference. The classification of the participants has been clarified in this line and more information has been added. In the Introduction section, the Diagnostic Criteria for Temporomandibular Disorders (DC/TMD) has been referenced.
In Methods section, the inclusion of participants according to DC/TMD criteria has been added, both in the text and in Table 1 (Participants inclusion and exclusion criteria).
In Results section the characteristics of the participants following the DC/TMD criteria has been added.
Manuscript:
- Lines 47 to 49 “Therefore, the term TMD is broad and it contains a number of disease entities. Thus, in order to support evaluation and diagnosis an assessment protocol, the Diagnostic Criteria for Temporomandibular Disorders (DC/TMD) has been developed.”
- Lines 84 and 85 “applying the DC/TMD,”
- Line 87, Table 1: “Diagnosed with pain disorders according to DC/TMD.”
- Line 186 and 187: “Participants were classified with myofascial pain (EG, n=4; CG, n=4) and myofascial pain with referral (EG, n=4; CG, n=4) according to DC/TMD.”
Point 3: Line 54-62: I would reformulate these sentences by focusing on the possible anatomical-functional relations linking TMD with the cervical area and then specify the effects of the manual therapy.
Response 3: It is indeed an important relation, and therefore, we have added in the Introduction some information regarding the link between TMD and the cervical area. Moreover, this is discussed in Lines 272 to 283.
Manuscript:
- Lines 62 to 64: “In this regard, a relationship between TMD and cervical dysfunction (one of the major goals of MT techniques) has been highlighted by several authors, explained by the topographic arrangement between the cervical spine and the trigeminal nerve.”
Point 4: Methods. 2.1 Participants - Table I- inclusion criteria- Joint noises during mouth opening. I would reformulate specifying the type classifying it according to the Dworkin classification (if possible).
Response 4: The type of noise, according to the DC/TMD has been added in the inclusion criteria information.
Manuscript:
- Line 87 Table 1 Participants inclusion and exclusion criteria.: “Occasional joint noises (clicks) during mouth opening”
Point 5: 2.4 Interventions. Personalized occlusal splint: I would only specify if there are any specific characteristics of the splint that according to the authors should be taken into consideration or not.
Response 5: We concur with the reviewer that splints are of common use, however, information related to the occlusal splint has been specified in relation to their composition in order to make the study procedure reproducible.
Point 6: Discussion. Considering the low impact of the present work as a preliminary study I would start the discussion reformulating the first sentence and making it “softer”.
Response 6: The initial sentence of the discussion has been reformulated as suggested by the reviewer.
Manuscript:
- Lines 254 to 257: “The current preliminary study shows that a MT-based protocol combined with ST tends to improve short- and midterm clinical parameters of pain (VAS and algometry), severity, dysfunction and perception of change in individuals with TMD when compared to the traditional treatment based on ST alone”
Point 7: Line 267: I would spend some more words explaining how the manual therapy can cause positive effects on pain, as here lies the core of the study.
Response 7: In order to highlight the positive effects of pain which, as the reviewer comments, is the core of the study, we have added more information in this regard.
Manuscript:
- Lines 285 to 288: “It appears that MT can improve pain due to the existence of an intimate functional relationship between the mandibular and the head‐neck systems. Our study provides preliminary evidence that MT applied both at the cervical spine and the masticatory area may be beneficial in decreasing pain in patients suffering from TMD.”
Point 8: Line 282: How do the author can explain why in the CG has been reported “minimally worse” in 50% and “no change” in 25% of the patients?
Response 8: The changes in the CG from feeling “minimally worse” (50%) and “no change” (25%) may be due to the Hawthorne Effect, by which participants show a change in some aspect of their behaviour as a result of knowing that they are being studied and not in response to any type of manipulation contemplated in the study. In this case, since they could have felt only receiving a conventional therapy, and no technique was applied on them directly, they may have felt “not improving”. (McCarney R, Warner J, Iliffe S, van Haselen R, Griffin M, Fisher P (2007). «The Hawthorne Effect: a randomised, controlled trial». BMC Med Res Methodol. 7: 30. PMC 1936999. PMID 17608932. doi:10.1186/1471-2288-7-30.)
Point 9: Conclusions. As for the first paragraph of the Discussion section I would suggest the authors to reconsider the conclusion in a “softer way”.
Response 9: We concur, and have written the Conclusions in a “softer way” as suggested by Reviewer.
Manuscript:
- Lines 351 to 355: “The results of the present study suggest that a combined protocol based on MT plus ST tend to improve pain, pain-induced disability and the patients’ self-perception of change in individuals with TMD. Moreover, such positive effects are maintained after one-month follow-up. These results reinforce the effectiveness of a multimodal treatment provided by a multidisciplinary team collaborating in the therapeutic approach for TMD patients.”
Point 10: Tables and Figures: I found all the tables clear and well structured. I would suggest the authors to pay attention in Table 3 and 4 there are still some “,” instead of “.”. I personally found the Appendix interesting as additional value to the study.
Response 10: Thank you for the comment. Changes have been applied in the Tables.

Round 2
Reviewer 1 Report
This article investigating the efficacy of combining splint therapy with manual therapy for the treatment of TMD has been significantly improved.
However, there are still unclear points concerning the definition of the study group and inclusion and exclusion criteria.
The authors state the inclusion criteria as below,
- Aged 18 to 65.
- Diagnosed with myogenic TMD or myofascial pain according to Helkimo Index.
- Diagnosed with pain disorders according to DC/TMD.
- Having one or more of the following symptoms:
- Deviation during active mouth opening (<40 mm).
- Pain in temporal and masseter region.
- Muscle sensitivity in at least one trigger point in the masseter and temporal muscles.
- Occasional joint noises (clicks) during mouth opening.
Does this mean the subject had to meet all criteria to be included in the study?
In that case what was the criteria for the Helkimo index to be diagnosed as myogenic TMD?
Also, if the patient had only joint noises during mouth opening how could this patient still be grouped as a muscle pain patient.
In the exclusion criteria, the authors state that “Presence of other orofacial or TMJ disk disorders” was a reason for exclusion. If the patient has deviation on opening this suggests the possibility of a TMJ disc disorder.
In the same line, the authors state in the Results that “Participants were classified with myofascial pain 186 (EG, n=4; CG, n=4) and myofascial pain with referral (EG, n=4; CG, n=4) according to DC/TMD”.
If the patient had both joint noises and muscle pain, the patient should be given a dual diagnosis according to DC/TMD.
Please clarify theses points.
Author Response
Response to Reviewer 1 Comments, Round 2
In relation to the manuscript with Title “Effect of manual therapy and splint therapy in people with temporomandibular disorders: a preliminary study” we would like to thank the reviewer for the comments on our manuscript
We concur with the reviewer that the information previously given may have misled in the understanding of the inclusion and exclusion criteria. We thank the reviewer for rising this issue, since it is of paramount importance in the understanding of the study.
We would like to clarify that participants were included in regard to the first three criteria listed (Aged 18 to 65; Diagnosed with at least mild TMD signs and symptoms according to Helkimo Index; Diagnosed with pain disorders according to DC/TMD) and in relation to the last criterion only “one or more” of the listed symptoms were needed for the participant to be included. When we edited the information of the table it seemed all the symptoms were included, but this was not the case. We should have edited the symptoms as a sublist in order to offer clear information:
- “Having one or more of the following symptoms:
- Deviation during active mouth opening (<40 mm).
- Pain in temporal and masseter region.
- Muscle sensitivity in at least one trigger point in the masseter and temporal muscles.
- Occasional joint noises (clicks) during mouth opening”
Therefore, although some patients could have had deviation of active mouth opening or occasional joint noises, this was not compulsory. In the studied participants none had these symptoms and, thus they were classified as “myofascial pain” and “myofascial pain with referral”.
Although the main objective of the study was to assess pain, we thought it could be interesting to include some patients that may have had occasionally some mild joint disorder symptoms, but we did not include those that had “Presence of other orofacial or TMJ disk disorders” diagnosed by dentists.
Having said all this, and in order to clarify our methods, overall the information as was written initially in the manuscript was misleading, plus we aim to concur with the standardised consensus about the evaluation of TMD. Therefore, we think it is appropriate to identify our inclusion criteria in relation to the symptoms only by the criterion of “pain disorders according to DC/TMD”. We think that, if the reviewer considers it appropriate, the last criterion that was shown in the table should be eliminated so the information is more clear.
Therefore, since the comments are all related to the definition of the study group and inclusion and exclusion criteria, changes have been applied in Table 1 (Line 87) as shown highlighted in yellow in the manuscript. However, responses have been addressed in different points in order to be able to clarify all information.
We apologize for the inconvenience and misleading of the initial information.
Point 1: This article investigating the efficacy of combining splint therapy with manual therapy for the treatment of TMD has been significantly improved. However, there are still unclear points concerning the definition of the study group and inclusion and exclusion criteria.
The authors state the inclusion criteria as below,
- Aged 18 to 65.
- Diagnosed with myogenic TMD or myofascial pain according to Helkimo Index.
- Diagnosed with pain disorders according to DC/TMD.
- Having one or more of the following symptoms:
- Deviation during active mouth opening (<40 mm).
- Pain in temporal and masseter region.
- Muscle sensitivity in at least one trigger point in the masseter and temporal muscles.
- Occasional joint noises (clicks) during mouth opening.
Does this mean the subject had to meet all criteria to be included in the study?
Response 1: We concur with the reviewer that the information given previously may have mislead in the understanding of the inclusion criteria. Participants did not have to meet all the criteria, only the first three. In relation to the last criterion only one of the sublist was needed for the participant to be included. Thanks to the reviewer’s comments, we have understood that the information in relation to the symptoms may have misled, and therefore the information of the Table has been changed.
Manuscript: Line 87, Table 1.
Point 2: In that case what was the criteria for the Helkimo index to be diagnosed as myogenic TMD?
Response 2: We would like to thank the reviewer for this comment since the Helkimo Index classifies dysfunction of TMD, and therefore the information given was not accurate. We have changed this criterion to clarify information. Participants were included if they had at least mild TMD signs and symptoms in regard to the Helkimo Index.
Manuscript: Line 87, Table 1.
Point 3: Also, if the patient had only joint noises during mouth opening how could this patient still be grouped as a muscle pain patient.
Response 3: As stated before, in the design of the study participants were included if they referred occasional joint noises during mouth opening, that is if “at some point” they had felt clicks, but as this is not accurate enough and no patients referred it the criteria has been clarified in relation to the DC/TMD.
Manuscript: Line 87, Table 1.
Point 4: In the exclusion criteria, the authors state that “Presence of other orofacial or TMJ disk disorders” was a reason for exclusion. If the patient has deviation on opening this suggests the possibility of a TMJ disc disorder.
Point 5: In the same line, the authors state in the Results that “Participants were classified with myofascial pain 186 (EG, n=4; CG, n=4) and myofascial pain with referral (EG, n=4; CG, n=4) according to DC/TMD”.
If the patient had both joint noises and muscle pain, the patient should be given a dual diagnosis according to DC/TMD.
Response 4 and 5: We concur with the reviewer that the information given previously in the manuscript was not clear in this regard. We have changed the given information to clarify the inclusion criteria and in regard to the exclusion criteria the word “presence” has been changed to “diagnose” in relation to TMJ disk disorders to make it more clear.
Manuscript: Line 87, Table 1.

Reviewer 2 Report
I would like to thank authors for their effort in increasing the pertinence and interest of their article
Author Response
Response to Reviewer 2 Comments
In relation to the manuscript with Title “Effect of manual therapy and splint therapy in people with temporomandibular disorders: a preliminary study” we would like to thank the reviewer for the comments on our manuscript all through the reviewing process.